# Delayed Initiation of Breastfeeding and Role of Mode and Place of Childbirth: Evidence from Health Surveys in 58 Low- and Middle- Income Countries (2012–2017)

**DOI:** 10.3390/ijerph18115976

**Published:** 2021-06-02

**Authors:** Shahreen Raihana, Ashraful Alam, Nina Chad, Tanvir M. Huda, Michael J. Dibley

**Affiliations:** 1Faculty of Medicine and Health, Sydney School of Public Health, The University of Sydney, Camperdown, NSW 2006, Australia; neeloy.alam@sydney.edu.au (A.A.); nina.chad@sydney.edu.au (N.C.); tanvir.huda@sydney.edu.au (T.M.H.); michael.dibley@sydney.edu.au (M.J.D.); 2Maternal and Child Health Division, International Centre for Diarrhoeal Disease Research, Bangladesh (icddr,b), Dhaka 1212, Bangladesh

**Keywords:** breastfeeding, initiation, caesarean section, vaginal births, health facilities, LMICs

## Abstract

Background: Timely initiation of breastfeeding is the first step towards achieving recommended breastfeeding behaviours. Delayed breastfeeding initiation harms neonatal health and survival, including infection associated neonatal mortality. Eighty percent of neonatal deaths occur in the low-and middle-income countries (LMICs), where delayed breastfeeding initiation is the highest. Place and mode of childbirth are important factors determining the time of initiation of breastfeeding. In this study, we report the prevalence of delayed breastfeeding initiation from 58 LMICs and investigate the relationship between place and mode of childbirth and delayed breastfeeding initiation in each country. Methods: We analysed data from the most recent Demographic and Health Survey (DHS) and Multiple Indicator Cluster Survey (MICS) collected between 2012 and 2017 and reported by 2019. The study sample comprised all women who had a live birth in the 24 months preceding the survey. ‘Delayed’ initiation of breastfeeding was defined using WHO recommendations as starting breastfeeding after one hour of birth. We coded the stratifying variable for the place and mode of childbirth as “vaginal birth at a facility (VBF)”, “caesarean section birth (CSB) “, and “vaginal birth at home (VBH)”. We used respondent-level sampling weights to account for individual surveys and de-normalised the standard survey weights to ensure the appropriate contribution of data from each country. We report the prevalence and population attributable fractions with robust standard errors. The population attributable risk identifies the proportion of delayed initiation that we could avert among VBH and CSB if everyone had the same risk of delaying breastfeeding as in VBF. Results: The overall prevalence of delayed initiation of breastfeeding was 53.8% (95% CI 53.3, 54.3), ranging from 15.0% (95% CI 13.8, 16.2) in Burundi to 83.4% (95% CI 80.6, 86.0) in Guinea. The prevalence of delayed initiation of breastfeeding was consistently high among women who experienced caesarean section births; however, there was no direct association with each country’s national caesarean section rates. The prevalence of delayed initiation among women who experienced VBF was high in Sub-Saharan Africa and South Asia, even though the CSB rates were low. In some countries, women who give birth vaginally in health facilities were more likely to delay breastfeeding initiation than women who did not. In many places, women who give birth by caesarean section were less likely to delay breastfeeding initiation. Population attributable risk percent for VBH ranged from −28.5% in Ukraine to 22.9% in Moldova, and for CSB, from 10.3% in Guinea to 54.8% in Burundi. On average, across all 58 countries, 24.4% of delayed initiation could be prevented if all women had the same risk of delaying breastfeeding initiation as in VBF. Discussion: In general, women who give birth in a health facility were less likely to experience delayed initiation of breastfeeding. Programs could avert much of the delayed breastfeeding initiation in LMICs if the prevalence of delayed initiation amongst women who experience CSB were the same as amongst women who experience VBF. Crucial reforms of health facilities are required to ensure early breastfeeding practices and to create pro-breastfeeding supportive environments as recommended in intervention packages like the Baby-friendly hospital initiative and Early essential newborn care. The findings from this study will guide program managers to identify countries at varying levels of preparedness to establish and maintain a breastfeeding-friendly environment at health facilities. Thus, governments should prioritise intervention strategies to improve coverage and settings surrounding early initiation of breastfeeding while considering the complex role of place and mode of childbirth.

## 1. Introduction

The World Health Organization (WHO) recommends all mothers be ‘supported to initiate breastfeeding soon after birth, ‘within the first hour after delivery’ [1]. The Baby-Friendly Hospital Initiative (BFHI) and Early Essential Newborn Care packages are simple, evidence-based interventions that emphasise the importance of initiating breastfeeding within the first hour of birth [2,3]. Breastfeeding all infants within the first hour of birth could prevent many newborn deaths [4,5,6] and early newborn illnesses [5,7]. 

In 2019, 47% of children under five years died in the neonatal period, three-quarters of whom died in the early newborn (0–7 days) stage [8,9]. Infections or sepsis are one of the leading causes of newborn deaths [10]. The majority of these neonatal deaths (80%) occurred in the low-and middle-income countries (LMICs) of Sub-Saharan Africa and Central and Southern Asia [8], which is also where there are reported low breastfeeding initiation rates at 47% in Sub-Saharan Africa and 39% in South Asia in 2015 [11]. The prevalence of delayed initiation of breastfeeding ranges from 5% to 86% across all countries [11]. 

Early initiation of breastfeeding influences breastfeeding success and positively impacts continuation into infancy [12,13,14] by stimulating a continuous production of breastmilk. Initiation of breastfeeding after the first hour is associated with severe illnesses [15], infection, and sepsis [16,17], and increased newborn and child mortality [4,5,6,18]. Early initiation of breastfeeding also reduces the likelihood of postpartum haemorrhage [19]. Delaying breastfeeding initiation beyond the first hour of birth is associated with an increased likelihood of introducing pre-lacteal feeds [20], i.e., giving any food or liquid other than breast milk before initiating breastfeeding. Delayed breastfeeding initiation also leads to a lower likelihood of introducing the first milk, colostrum, thus depriving the newborn of the antibodies and immunoglobulins present in it and increasing the risk of adverse outcomes, including sepsis and infection later in life [7,21,22]. A dose-response relationship also exists with an increasing risk of morality with greater delays in breastfeeding initiation beyond the first hour [23] till day seven [24]. Overall the risk of mortality is increased by 2.4 times if breastfeeding is initiated after the first day [24] compared to initiation within the first day of birth. A review of 18 studies [5] found delayed initiation of breastfeeding reduced the pooled risk of all-cause mortality by 44% among newborns who survived past 48 h after birth and by 42% among low birth weight infants. Initiating breastfeeding between 2 and 23 h after birth is associated with a 33% increased risk of neonatal death compared to breastfeeding within that first hour of birth [23,25].

Despite the improvements in other breastfeeding practices, there has been very limited progress with early breastfeeding initiation. Globally the rate of early or timely breastfeeding initiation is estimated to have increased only 14%, from 32% in 2000 to 46% in 2017 [26]. Literature from several countries, including Nigeria [4,27,28], Sri Lanka [29], Nepal [30], Ethiopia [31,32], Indonesia [33], Malawi [34], Uganda [35], and India [36], suggests that the place of birth, mode of birth and the skill level of the attendant present at birth are important determinants of early (or timely) initiation of breastfeeding. Most evidence suggests that early breastfeeding initiation is higher among women experiencing hospital births in LMICs [4,28,29,30,31,33,34]. However, the findings are not consistent for all hospital births. Studies in Nigeria [27], Ethiopia [32], Uganda [35], and India [36] report that mothers who experienced a caesarean section birth at a health facility had a significantly higher likelihood of delaying initiation of breastfeeding beyond the first hour of birth, compared to those experienced vaginal births. Understanding the predictors of delayed initiation of breastfeeding across comparable settings could help identify modifiable risk factors and facilitate improvement in EIBF practices.

Our study aims to describe the recent country-level prevalence of delayed breastfeeding initiation by place and mode of childbirth in 58 LMIC countries using publicly available survey data. While some studies have examined the multi-country prevalence of breastfeeding initiation rates [20,37], few papers have included standard, comparable community-level data sources from LMICs from around the world. Unlike previous studies, the unique aspect of our analysis is that we examine the proportion of delayed initiation of breastfeeding that could be averted if all women had the same risk of delaying initiation as those who experience a vaginal birth in a health facility. Findings from this study will suggest effective strategies that governments and program managers can prioritise to improve coverage and settings surrounding early initiation of breastfeeding and, in turn, increase accountability by appropriately using monitoring data to enhance the quality of care.

## 2. Materials and Methods

### 2.1. Design and Data Sources

This study analysed publicly available secondary data sources of nationally representative cross-sectional surveys, including Demographic and Health Surveys (DHS) and Multiple Indicator Cluster Surveys (MICS). The data are from low- and middle-income countries and were collected between 2012 and 2017 and reported before January 2020. The DHS and MICS follow a two-stage cluster random sampling design to select nationally representative samples of households from enumeration areas drawn from the country’s national censuses. Both surveys use standardised questionnaires and measurement techniques to collect information from women on their birth and reproductive history and outcomes, place of childbirth and practices immediately after delivery alongside socio-demographic characteristics at the individual, household and community level. The methodological details of both surveys are published elsewhere [38,39]. The uniformity of the information and variables collected in both surveys across all LMICs and corresponding survey waves makes it easy to compare results and indicators across countries. Both surveys are typically administered, conducted and implemented by national statistical agencies of respective countries. Both types of surveys administer a major part of the questionnaires to women of reproductive age (15 to 49 years). We identified LMICs from the World Bank list of economies, and income groups last updated in June 2018 and used 58 LMIC surveys (49 DHS and 9 MICS surveys) for the analyses. Both DHS and MICs use weighted samples to ensure the characteristics of the sample align with those of the population.

### 2.2. Study Sample

The study sample comprised of women who had a live birth in the two years preceding the survey. For the DHS, we extracted data from the ‘birth’ and ‘household’ record files; for MICS, data were from the ‘women’, ‘household’, and ‘child’ record files. 

### 2.3. Data

We excluded eight surveys from LMIC conducted between 2012 and 2017 based on the following criteria: (i) survey did not collect data for or report on the time of initiation of breastfeeding (*n* = 1, Colombia DHS 2015) (ii) datasets were not available as STATA files and were not able to be exported into STATA with appropriate labels (*n* = 2, Congo and El Salvador MICS 2014), (iii) surveys for which the authors’ calculated weighted population of women whose last child was born in the two years preceding the survey did not match the weighted population presented in the report (*n* = 2, Kenya DHS 2014 and Turkey DHS 2013), (iv) surveys without sample weights in the dataset (*n* = 1, Paraguay MICS 2016), (v) unique identifiers in the birth and household record files were not uniform, and the file records could not be merged (*n* = 2, Mexico MICS 2015 and Peru DHS 2014). Appendix A, Table A1 shows the list of surveys, the number of women interviewed, and the number of households sampled in each survey included for this study.

We cleaned the included survey data, consistently labelled variables, restricted it to women who had had a live birth in the two years preceding the survey and then appended them into a single data file for analysis. We created new variables to classify each country by World Bank region (East Asia and Pacific, Europe and Central Asia, Latin America and Caribbean, the Middle East and North Africa, South Asia, and Sub-Saharan Africa) [40]. Appendix A, Table A2 shows the general characteristics of the surveys, including survey year, world regions, and income groups distribution.

### 2.4. Outcome Variable

‘Delayed’ initiation of breastfeeding was the main outcome of interest. The primary source of breastfeeding initiation data was from the birth record file in the DHS dataset and the women record file in the MICS dataset. In both DHS and MICS, the data for the timing of initiation of breastfeeding was collected from mothers using the unprompted self-reported question ‘How long after birth did you first put (name of child) to the breast?’. We categorised it as a dichotomous variable of ‘Yes’ for women putting their child to the breast after the first hour of birth and ‘No’ for those who initiated breastfeeding within the first hour. We used the WHO recommended cut off for initiation of breastfeeding [1] within the first hour of birth.

### 2.5. Stratification Variables

The main stratifying variables used in this study are the mode of childbirth and the place of delivery. In both DHS and MICS data, we categorised the mode of birth as ‘vaginal birth’ (VB) or ‘caesarean section’ (CS). Data was collected using the self-reported question ‘Was (name) delivered by caesarean section?’. We coded respondents who said ‘no’ to this question as having had a vaginal birth. Both DHS and MICS questionnaires broadly classified the place of childbirth as (i) at ‘home’, including the woman’s place of residence or any other house, and (ii) at ‘health facility’, referring to the government and privately owned hospitals health centres or clinics. We classified both public and private sectors as ‘institutional’ or ‘health facility’ births [39]. We combined these two variables to construct a single variable for the place and mode of childbirth, coded as “vaginal birth at a health facility” (VBF), “caesarean section birth” (CSB), and “a birth at home” (VBH).

Background characteristics considered as covariates for this analysis included the mother’s age, place of residence, household wealth quintile, mother’s education, and perceived child size at birth. We also extracted national-level data for neonatal mortality rate (neonatal deaths per 1000 live births), the prevalence of CS, and the proportion of health facility births. These data provide important contextual information to support the interpretation of the analysis.

### 2.6. Statistical Analysis

We analysed country-level DHS and MICS data to present the proportion of delayed breastfeeding initiation among women aged 15–49 years with a live-born child in the two years preceding the survey. We also computed crude and adjusted population attributable fractions for delayed initiation of breastfeeding. We used respondent level sampling units to account for individual surveys. We used the survey weights already in the datasets to obtain all country-level estimates. The survey weights in DHS and MICS were normalised to make the total number of unweighted cases equal to the total number of weight cases at the national level, and this was a survey specific calculation. However, analysis of pooled data requires de-normalisation of the standard survey weights to ensure valid estimates of total numbers from each survey is included in the final dataset. For this purpose, we used the number of women 15–49 years in each of these countries as reported by the UNDP population survey to de-normalise the given standard weights [41]. 

We conducted a descriptive analysis for each country to estimate the proportion of urban dwellers, households in the lowest quintile, mothers with no education, newborns perceived to have smaller than the average size at birth, and the mean age of mothers among the ‘delayed’ and ‘early’ initiator groups. We estimated the country level proportion of delayed initiation among women who experienced VBF, VBH, or CSB. To examine the differences in prevalence between the early and delayed initiators, we then present the country level prevalence for delayed breastfeeding initiation separately among VBH, VBF, and CSB. We examined the country level risk of delayed initiation among the three strata by estimating the crude and adjusted population attributable risks.

We used modified Poisson regression to calculate robust estimates for the unadjusted and adjusted prevalence ratios for risk of delaying breastfeeding initiation in each stratum, with 95% confidence intervals. We then used the regression post estimation command ‘regpar’ to calculate the country level crude and adjusted population attributable fraction (PAF). We considered the risk of delayed breastfeeding initiation among VBF as the reference to calculate the population attributable fractions for VBH and CSB. Here, the PAF describes the proportion of delayed breastfeeding initiation that programs could ideally avert among women who experienced VBH and the CSB if all women had the same risk of delaying initiation as in VBF.
$$Population\;Attributable\;Fraction=P_{b}\left(\frac{PR-1}{PR}\right)$$
Here, *P_b_* is the proportion of delayed initiators among women who experienced VBH or CSB. The *PR* is the prevalence ratio of delayed initiation among women who experienced VBH or CSB with reference to women who experienced VBF. For ease of interpretation, we presented the PAF as the population attribution risk percent (PAR%) by multiplying PAF by 100. We calculated the ‘crude PAR%’ and the ‘adjusted PAR%’ to present the proportion of delayed initiation that could be averted among VBH and CSB if everyone had the same risk of delaying as in VBF. We estimated 95% CI for all PAR%. We performed all statistical analyses using STATA version 15 (Stata Corporation, College Station, TX, USA).

## 3. Results

The study included 298,656 women from 58 countries who gave birth in the two years preceding the DHS and MICS surveys between 2012 and 2017. Table 1 shows the country level weighted numbers and background characteristics of women aged 15–49 years with a live birth in the two years preceding the survey, stratified as ‘early’ and ‘delayed’ initiators. The mean age of survey respondents ranged from 31.3 years in Tunisia to 24.2 years in Bangladesh among the delayed initiators and 30.9 years in Tunisia to 23.9 years in Bangladesh among the early initiators. The data included 24,022 women aged 15–19 years, 1,77,951 women aged 20–29, and 96,683 women aged 30 years or older. In all countries, the overall median time to initiate breastfeeding, among those who initiated after the first hour of birth, was 2 hours (interquartile range was 47 h). Among the delayed initiators, the percentage of pregnant women with no formal education ranged from 0.0% in Armenia, Kyrgyz Republic and Ukraine to 87.8% in Niger. While among the early initiators, the percentage of pregnant women with no formal education ranged from 0.0% in Armenia and the Kyrgyz Republic to 83% in Niger. Among delayed initiators, the percentage of lowest wealth quintile households ranged from 11.4% in Thailand to 26.9% in Sudan, and among early initiators from 14.1% in Moldova to 34.6% in Guyana. Among the delayed initiators, the percentage of newborns perceived by mothers to have smaller than the average size at birth range from 10.4% in Timor Leste to 40.7% in Yemen and among early initiators from 4.2% in Ukraine to 34.6% in Sudan.

Table 1 and Appendix A, Figure A1 present the country-specific weighted percentage of women who initiated breastfeeding after the recommended first hour of birth. There was no breastfeeding initiation within an hour of birth in 20 countries (34.5%) for more than half the newborns. The regional level percentage of women who delayed breastfeeding initiation was greater than 50% in South Asia and the Middle East, and North Africa, with South Asia having the highest percentage of delayed initiation at 58.6%. The highest percentage (55.6%) of delayed initiators was among women in countries in the lower-middle-income group. In comparison, countries in the low and upper-middle-income groups had a lower overall percentage of women who delayed breastfeeding initiation (44.9%).

Seventeen percent of women experienced caesarean section births (Table 2). Regionally, the highest prevalence of CSB was in the Middle East and North Africa (43.1%), followed by Latin America and the Caribbean (31.0%), East Asia and Pacific (20.2%) and South Asia (20.0%). The percentage of women who experienced CSB ranged from 1.4% in Niger to 60.6% in the Dominican Republic. 

Women who experienced CSB were more likely to delay breastfeeding initiation. In 51 of the included countries, more than half the women who experienced CSB had breastfeeding delayed beyond one hour of birth. In 28 countries, more than 50% of women who experienced VBH delayed breastfeeding initiation. In 17 countries, more than half the women who experienced VBF delayed breastfeeding initiation beyond the first hour. We noted substantial variation in delayed initiation from 11.7% and 15.8% in Burundi to 80.8% and 84.7% in Guinea amongst women who had given birth vaginally (facility and home births). Figure 1 shows the prevalence of delayed initiation of breastfeeding among the three groups for the place and mode of childbirth in a hundred per cent stacked bars to present the relative difference among delayed initiators in each country by World Bank regions. Across the countries, the prevalence of delayed initiation of breastfeeding was lowest amongst women who had given birth vaginally in a health facility and highest amongst women who had experienced caesarean section births. However, the relative contribution of mode and place of childbirth among all delayed initiators differed between countries (Figure 1). The highest prevalence of delayed initiation of breastfeeding occurred amongst VBF in Sao Tome and lowest amongst VBH in Armenia, Moldova, Albania, the Dominican Republic, and Jordan.

Table 3 presents the country level crude and adjusted PAR% for delayed breastfeeding initiation among VBH and CSB relative to VBF. The reference group for calculating the PAR% is children born through vaginal birth at a health facility. These results indicate the percentage of delayed breastfeeding initiation potentially averted if all women in each country experienced VBF. For VBH, the PAR% ranged from −33.1% in the Maldives to 76.8% in Kazakhstan. For CSB, the crude PAR% ranged from 1.4% in the Maldives to 52.9% in Burundi. When adjusted for several known covariates, the adjusted PAR% for VBH range from −28.5% in Ukraine to 22.9% in Moldova. The adjusted PAR% for CSB ranged from 10.3% in Guinea to 54.8% in Burundi. We could not calculate an adjusted PAR% for the Philippines. Yemen and the Maldives as an adjusted Poisson model could not converge as estimates for some adjusting variables do not exist in the dataset. The negative values of PAR% for VBH suggests that even if all women in those countries experienced VBF, it would still not avert delayed breastfeeding initiation. In countries with a negative value for PAR%, delayed breastfeeding initiation was mostly higher among vaginal birth in the health facilities compared to vaginal births at home.

## 4. Discussion

The relation between delayed initiation of breastfeeding beyond the first hour of birth and the place and mode of childbirth is not consistent across contexts. In 26 countries, VBF was associated with an increased risk of delayed breastfeeding initiation, and women who have experienced VBH were least likely to delay breastfeeding initiation. In all countries, CSB was associated with the highest risk of delayed initiation even though its overall contribution to the prevalence of delayed initiation was lower in countries where CSB is not much prevalent. Our findings suggest that programs promoting health facility-based births should also include evidence-based care and systems to support appropriate breastfeeding practices immediately after childbirth. They indicate the need for a holistic approach for institutionalising deliveries that combines health system strengthening and promotion and support of appropriate breastfeeding practices. Timely initiation of breastfeeding is vital to intervention packages like BFHI and EENC [2].

Our population attributable risk analysis suggests that improving breastfeeding practices for vaginal births at a health facility is not an effective solution in all settings. It is important to ensure appropriate breastfeeding support that considers the local environment, the health system capacity and the common feeding practices in each country. Our study reiterates that the prevalence of delayed initiation of breastfeeding is generally higher among women who had experienced a CSB compared to women who experienced VBF. It also highlights the need for program managers to understand the complex role of the mode and place of childbirth and design a range of country-specific interventions to create appropriate pro-breastfeeding environments around the time of birth.

This study has collated data from several countries of varying socio-economic status to examine the impact of the place and mode of childbirth, explore the complexity of the role of settings around birth, and present how responses may need to vary in different environments. We specifically focused on the distribution of the time of breastfeeding initiation across LMICs from all regions, and we compared variation in the timing of breastfeeding initiation in different delivery care settings. We have used the complex role of the mode and place of childbirth to present a composite stratification by different settings. Our results will help program managers and governments identify settings with varied preparedness to adhere to timely breastfeeding initiation following vaginal or caesarean section births at home or a health facility.

Our study has some limitations to consider when interpreting the findings. Firstly, we could not capture the country level variations due to cultural beliefs and norms that lead to variations in breastfeeding initiation time. Secondly, we could not capture the prevalence of preterm births and birthweight data as DHS/MICS surveys do not collect these objective data in all countries. Delayed initiation among term infants is more of a concern for newborns with very low birth weight and preterm infants than normal birth weight and term infants. This concern is mostly because low birth weight and preterm newborns are more likely to be unstable following birth [42] and require special newborn care to manage complications. 

Like our findings, Oakley et al. suggested that breastfeeding initiation is more dependent on ‘favourable’ childbirth settings [20]. In Sub-Saharan African countries like Ethiopia [43], one study reported that institutional birth increased the likelihood of delayed initiation compared to home birth. Another study in Pakistan [44] found that overall breastfeeding initiation and continuation practices were lower for births at a health facility, regardless of the mode of childbirth. It seems that for some countries, mostly those in Sub-Saharan Africa and South Asia, increased contact with health facilities at the time of birth is a risk factor for delayed initiation of breastfeeding. Such findings suggest that while institutional deliveries are being promoted and encouraged in most LMICs to ensure safe childbirth and post-birth care, there appears a gap in monitoring and implementing breastfeeding-related health care services. Such gaps may have resulted in VBH being protective for delayed breastfeeding initiation in some countries where the health system and the healthcare providers were not adequately equipped to create an appropriate environment for breastfeeding initiation following a VBF.

We found substantial variation in the prevalence of delayed initiation across the included countries. Delayed initiation is more than 50% for a third of the countries in Sub-Saharan Africa and a half of South Asia and the Middle East and North African countries. The prevalence of delayed breastfeeding initiation was lower among the delayed initiators in the lowest household wealth quintiles compared to the early initiators. This finding suggests that even though women from the poorest households are more likely to deliver at home and have limited contact with health professionals, it does not always interfere with appropriate early feeding practices. A large proportion of the delayed initiators perceived their newborns to have been smaller than the average size at birth. There are similar findings noted in studies in several other LMICs [4,45]. A potential reason for this is that the smaller than average-sized newborns are not physically mature to have breast-seeking reflexes, are unable to suckle, or health professionals are likely to intervene and separate mother and infant [30,46]. Even in countries with less than 30% delayed breastfeeding initiation, tailored interventions are needed to improve adherence to recommended breastfeeding initiation time by designing programs at the health service delivery platforms where most childbirths occur. Furthermore, it is important to consider the maternal and health facility level characteristics that are associated with breastfeeding initiation around the time of childbirth [15]. 

We found a higher prevalence of delayed initiation in all women who had experienced a caesarean section birth than vaginal births at home or a health facility. In 51 of the 58 countries, the delayed initiation among women who experienced CSB was higher than 50%, with the highest being in the Middle East and North Africa. In 10 of these 51 countries, CSB was higher than 20% of all births, and none were from Sub-Saharan Africa. This finding indicates that in most countries with a high proportion of delayed initiation among CSB, the prevalence of CSB is lower than 20%. Regardless of the lower CSB rates, the health systems were not well equipped to provide appropriate breastfeeding counselling and care following a caesarean section birth. 

In many settings, including Sub-Saharan African countries, CSB is a common risk factor for delayed breastfeeding initiation; however, the overall CSB rates are still low. Any program/intervention approach to improve the early initiation of breastfeeding in these countries with lower CSB rates needs to look at the distribution of delayed initiation among the vaginal births at home and at a health facility. In 12 of the 41 countries with CSB rates lower than 20%, delayed initiation among vaginal births at a health facility was higher than 50%. This finding reiterates that delayed initiation is often not just influenced by home birth settings, as some earlier studies have reported.

The population attributable fractions also indicate that vaginal births at home may be protective in some settings, suggesting a huge gap in breastfeeding-related services provided at health facilities. Thus, we should not consider the mode and place of childbirth separately to explain delayed initiation in many countries, particularly those in Sub-Saharan Africa. In this current study, we present a combined effect of the two attributes on delayed breastfeeding initiation.

In this study, we presented objective measures for the effect of the delay in breastfeeding initiation in a hypothetical setting, where all women experience vaginal birth at a health facility. Overall, it is clear from the crude and adjusted PAR% that in almost all countries included in the study, it is possible to avert delayed breastfeeding initiation if women experiencing CSB had the same risk of delaying initiation as women experiencing VBF. This finding indicates that countries with a high prevalence of CSB require crucial reforms of health facilities to ensure a pro-breastfeeding supportive environment regardless of the mode of childbirth. On average, 24.4% of the delayed initiation could be prevented in these 58 countries if women experiencing CSB had the same risk of delaying breastfeeding initiation as women experiencing VBF. This finding is more relevant to countries like the Dominican Republic, Thailand, Albania, Guatemala, Jordan, Tunisia, Bangladesh, and Armenia. Both the prevalence of CSB and the adjusted population attributable risk percent were more than 20%. In these countries, interventions to improve post-birth breastfeeding practices could significantly prevent adverse health outcomes in newborns experiencing delayed initiation. 

The negative PAR% for vaginal births at home implicates that moving all childbirths to a health facility setting would not prevent the delayed initiation. Several factors may be a likely explanation for this phenomenon. Firstly, the health system of the country and the health facility settings are not well equipped to create a favourable environment for breastfeeding initiation. Secondly, while equipped to undertake appropriate measures for childbirth, the healthcare providers could not provide the physical and psychological support for women to initiate breastfeeding on time. Thirdly, the proportion of births occurring at the health facilities may have been high (for example, in Gabon, Sao Tome, Guyana, and the Dominican Republic), indicating a burden on an unprepared health system. In these settings, it would not be possible to prioritise breastfeeding initiation regardless of the mode of childbirth at a health facility. Fourthly, the attitude and practice around caesarean section births among mothers and healthcare staff generally are that the mother would not easily breastfeed considering her post-operative medical condition. Thus the perceived need for artificial or formula feeding, which then delays breastfeeding initiation. 

In 2017, the WHO published a guideline [47] and implementation guidance [3] for early breastfeeding initiation for all settings, including in post-caesarean section births. The recommendation states that mothers who undergo a medical procedure during childbirth and are unable to breastfeed must have their newborn put to their breast as soon as she is conscious. In such circumstances, health care providers present at the birth should support mothers. The BFHI minimum requirements elaborate [47] that if a mother has not had general anaesthesia, the baby should be on her chest in skin-to-skin contact, no later than 10 minutes arrival in recovery unless the mother or the baby’s medical condition prevent this contact. In circumstances when the mother has had general anaesthesia during caesarean section birth, the baby should be on skin-to-skin contact “within 10 minutes of being able to respond to the baby” unless otherwise medically warranted. In all circumstances, such skin-to-skin contact is to be continued uninterrupted until after the first breastfeed or for at least an hour, unless the baby is fed sooner. In addition, countries (like the USA and Brazil) have implemented monitoring tools to assess attitudes about baby-focused care [48] and training courses on breastfeeding counselling [49]. Implementation involving scaling up utilisation of contextually appropriate adapted versions of such tools could be beneficial in monitoring pro-breastfeeding environment in resource-limited settings. 

Consistent with our study findings, several studies from low- and middle-income countries have linked caesarean section births with delayed breastfeeding initiation [50,51,52]. Most of these studies [51,52] and several other studies [53,54,55] also reported caesarean birth at a health facility to be a risk factor for delayed initiation of breastfeeding. However, in countries with relatively well-developed delivery care platforms and breastfeeding friendly environments at health facilities, studies have reported births at a health facility to be significantly protective [56,57], even in settings with high caesarean section rates [58]. Although few studies have specifically looked at delayed breastfeeding initiation among vaginal births at a health facility, those that have [52,59] found it a protective factor. In this study, vaginal birth at a health facility setting has proven to be both a risk and a protective factor in different country settings. Therefore, the health and nutrition program managers in each country must understand and appreciate that a single blanket approach that has been effective in one LMIC may not be effective in all settings. Donors should support governments to enhance platforms best suited for breastfeeding counselling and a supportive environment immediately following childbirth. Pro-breastfeeding advocates must also improve the quality of breastfeeding-related care immediately following birth in a health facility setting, regardless of the mode of delivery, but in line with the WHO recommended guidelines. Implementing plans must create a favourable post-childbirth breastfeeding initiation environment by focusing on developing human resources present at childbirth. The programs should raise awareness around the recommended guidelines following birth and train all staff and healthcare personnel around the importance of initiating breastfeeding within the first hour regardless of the mode of childbirth.

## 5. Conclusions

In many low and middle-income countries, increased contact with health facilities during childbirth can be a risk factor for delayed breastfeeding initiation. Institutionalising all childbirths needs to be accompanied by ensuring that the health system can promote and support appropriate breastfeeding practices regardless of the mode of childbirth. Even though BFHI and WHO guidelines promote the steps for initiating successful breastfeeding irrespective of the place and mode of birth, current essential newborn care practices in most of these LMICs do not align with the recommendations. Despite the availability of tools and recommendations for minimum care standards, there is a need for further investigations to identify and address barriers to quality health service delivery in many LMICs. It is crucial for maternal and child health program managers to design interventions adopting a breastfeeding friendly policy for institutional and home births. Adopting breastfeeding-friendly policies at scale and utilising monitoring data to increase accountability and improve service delivery can significantly reduce delayed breastfeeding initiation.

## Figures and Tables

**Figure 1 ijerph-18-05976-f001:**
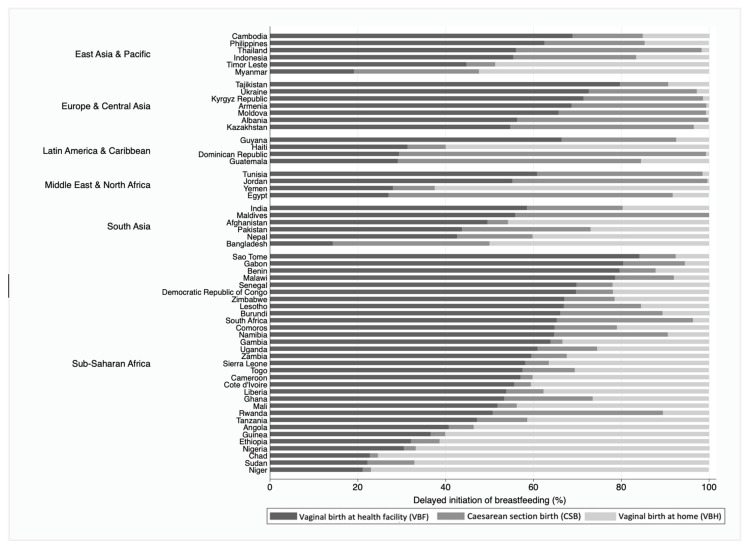
Mode of childbirth among the delayed initiators in each country.

**Table 1 ijerph-18-05976-t001:** Background characteristics among delayed initiators, i.e., women aged 15–49 years with a live-born child in the two years preceding the survey who delayed breastfeeding initiation beyond the first hour of birth.

Region/Country	Total Number Analysed (*n*)	Delayed Initiators (*n*)	Prevalence of Delayed Initiation of Breastfeeding (%)	Mean Age of Women (Years)	Urban Dwellers (%)	Lowest Household Wealth Quintile (%)	No Education (%)	Smaller than the Average Size of a Child at Birth (%)
East Asia & Pacific								
Thailand	2092	1257	60.1 (55.6, 64.4)	28.1 (±4.7)	46.7 (40.2, 53.3)	11.4 (8.9,14.6)	5.2 (2.8, 9.3)	11.6 (8.8, 15.1)
Indonesia	6616	2871	43.4 (41.7, 45.1)	29.5 (±3.7)	48.4 (45.1, 51.7)	20.9 (18.9,22.9)	0.9 (0.5, 1.4)	14.4 (13.0, 15.9)
Philippines	3725	1605	43.1 (40.4, 45.9)	28.1 (±4.2)	45.9 (40.1, 51.9)	17.8 (14.8,21.2)	1.2 (0.7, 2.2)	-
Cambodia	2944	1101	37.4 (34.8, 40.2)	27.3 (±8.8)	18.5 (14.5, 23.4)	19.5 (16.4,23.1)	13.8 (11.2, 16.9)	13.4 (11.1, 16.2)
Myanmar	1669	554	33.2 (30.1, 36.5)	29.5 (±4.2)	22.8 (17.4, 29.3)	21.2 (17.2,25.8)	16.0 (12.6, 20.1)	17.2 (14.0, 20.9)
Timor Leste	2866	711	24.8 (22.2, 27.5)	28.2 (±33.3)	29.1 (22.2, 37.2)	24.1 (19.6,29.2)	20.7 (17.6, 24.2)	10.4 (7.5, 14.1)
Europe and Central Asia								
Armenia	666	394	59.1 (54.1, 63.9)	27.5 (±8.8)	64.2 (55.1, 72.3)	21.0 (14.5,29.4)	6.4 (3.7, 11)	11.4 (8.7, 14.9)
Albania	1035	450	43.5 (38.7, 48.4)	28.4 (±12.1)	50.9 (43.2, 58.6)	16.4 (12.0,22.1)	0.6 (0.2, 2.1)	11.2 (8.1, 15.3)
Moldova	750	293	39.1 (35.0, 43.3)	27.2 (±8.4)	40.9 (33.9, 48.3)	16.9 (12.0,23.3)	0.9 (0.1, 5.9)	15.8 (11.5, 21.5)
Tajikistan	2481	953	38.4 (35.0, 41.9)	26 (±8.6)	18.5 (14.2, 23.8)	17.1 (13.6,21.3)	1.8 (1.0, 3.0)	14.6 (12.1, 17.5)
Ukraine	707	243	34.3 (30.5, 38.4)	27.2 (±3.5)	69.4 (61.4, 76.3)	17.8 (13.3,23.5)	0.0 (0.0, 0.0)	15.3 (10.5, 21.6)
Kazakhstan	2157	360	16.7 (14.7, 18.9)	28 (±7.9)	50.8 (42.1, 59.5)	16.4 (11.8,22.4)	5.5 (3.5, 8.7)	19.9 (15.2, 25.7)
Kyrgyz Republic	1696	275	16.2 (14.0, 18.6)	27.4 (±11.6)	34.0 (25.1, 44.1)	16.3 (10.4,24.5)	0.8 (0.1, 5.4)	23.8 (18.6, 29.9)
Latin America and the Caribbean								
Dominican Republic	1395	792	56.8 (52.5, 61.1)	25.9 (±7.6)	74.2 (64.6, 81.9)	17.0 (13.5,21.2)	1.4 (0.8, 2.4)	21.6 (17.4, 26.5)
Haiti	2424	1275	52.6 (50.1, 55.2)	27.9 (±11.1)	35.8 (30.0, 42.1)	19.3 (16.1,22.9)	15.7 (13.0, 18.7)	31.1 (28.0, 34.4)
Guyana	769	391	50.8 (47.0, 54.6)	27.1 (±28.7)	26.9 (20.3, 34.7)	24.5 (19.5,30.3)	2.9 (1.5, 5.4)	22.7 (18.5, 27.5)
Guatemala	4790	1768	36.9 (34.9, 38.9)	26.1 (±11.8)	44.1 (39.3, 48.9)	13.5 (11.3,16.0)	11.9 (10.1, 14.1)	21.5 (19.4, 23.7)
The Middle East and North Africa								
Egypt	6297	4591	72.9 (71.2, 74.5)	27.3 (±4.9)	31.8 (28.0, 35.8)	19.5 (17.4, 21.8)	15.0 (13.6, 16.6)	18.0 (16.5, 19.5)
Tunisia	1164	700	60.1 (56.2, 63.8)	31.3 (±5.7)	67.1 (60.0, 73.4)	18.0 (14.5, 22.1)	9.5 (7.3, 12.2)	17.8 (14.6, 21.4)
Yemen	6110	2890	47.3 (45.2, 49.5)	27.6 (±10.6)	24.6 (20.1, 29.8)	19.2 (16.1, 22.6)	-	40.7 (38.1, 43.3)
Jordan	3472	1146	33 (30.4, 35.6)	29.2 (±14.3)	87.0 (83.3, 89.9)	18.3 (14.5, 23.0)	0.9 (0.5, 1.5)	16.7 (14.0, 19.9)
South Asia								
Pakistan	3935	3164	80.4 (77.9, 82.7)	27.7 (±2.6)	32.4 (26.7, 38.7)	20.2 (16.2, 25.0)	46.2 (41.7, 50.7)	23.6 (21.3, 26.2)
Afghanistan	11,539	6820	59.1 (56.6, 61.6)	27.3 (±11.9)	25.5 (20.0, 32.0)	19.3 (16.4, 22.7)	81.5 (78.4, 84.2)	28.2 (25.3, 31.2)
India	94,111	55055	58.5 (58.0, 59.0)	25.6 (±4.5)	26.8 (26.2, 27.3)	20.3 (19.8, 20.8)	29.7 (29.1, 30.2)	13.6 (13.2, 14.0)
Bangladesh	3205	1577	49.2 (46.5, 51.9)	24.2 (±2.7)	29.0 (23.9, 34.7)	18.4 (15.2, 22.2)	12.2 (10.0, 14.9)	21.2 (18.7, 23.8)
Nepal	1978	892	45.1 (42.0, 48.1)	25 (±4.8)	51.2 (43.4, 58.9)	16.3 (12.9, 20.4)	32.6 (27.9, 37.5)	18.7 (15.9, 21.8)
The Maldives	1086	364	33.5 (29.5, 37.8)	29.1 (±26.1)	48.3 (38.1, 58.6)	23.1 (17.2, 30.1)	0.8 (0.3, 2.1)	-
Sub-Saharan Africa								
Guinea	2818	2350	83.4 (80.6, 86.0)	27.2 (±12.1)	25.9 (20.6, 31.9)	20.2 (16.8, 24.2)	75.4 (72.9, 77.8)	13.4 (11.7, 15.3)
Chad	6742	5191	77.0 (74.6, 79.2)	26.8 (±14.9)	19.9 (16.2, 24.2)	20.2 (18.1, 22.6)	59.3 (56.9, 61.7)	23.7 (21.8, 25.7)
Cote d’Ivoire	3039	2103	69.2 (65.7, 72.5)	27.2 (±8.5)	37.5 (31.0, 44.4)	20.2 (16.8, 24.0)	62.0 (58.4, 65.4)	16.9 (14.8, 19.3)
Cameroon	2977	2048	68.8 (66.1, 71.4)	27.1 (±7.7)	39.4 (33.8, 45.4)	26.8 (21.9, 32.4)	31.9 (28.4, 35.6)	18.5 (16.2, 20.9)
Gabon	2102	1423	67.7 (63.5, 71.5)	26.8 (±26.3)	84.2 (79.1, 88.3)	19.3 (15.5, 23.8)	5.7 (3.7, 8.7)	19.7 (17.0, 22.8)
Nigeria	12,473	8332	66.8 (65.0, 68.6)	28 (±6)	31.7 (28.0, 35.6)	20.9 (18.5, 23.6)	50.8 (48.3, 53.4)	16.9 (15.6, 18.2)
Senegal	4447	2953	66.4 (64.2, 68.6)	28.4 (±12.9)	36.7 (30.9, 42.9)	19.0 (16.1, 22.4)	61.6 (59.0, 64.1)	36.8 (34.5, 39.2)
Comoros	1298	861	66.3 (62.1, 70.2)	28.3 (±29.1)	30.5 (23.5, 38.4)	20.3 (16.1, 25.3)	42.3 (38.2, 46.6)	27.9 (24.1, 32.1)
Sao Tome	756	466	61.7 (56.5, 66.7)	27.9 (±45.2)	68.7 (57.8, 77.8)	17.8 (13.8, 22.7)	3.2 (1.8, 5.7)	15.0 (11.5, 19.3)
Angola	5405	2794	51.7 (48.9, 54.4)	26.8 (±10.9)	58.3 (52.4, 64.0)	19.5 (16.7, 22.7)	28.8 (26.1, 31.7)	12.0 (10.2, 14.0)
Tanzania	4076	1985	48.7 (46.0, 51.4)	27.6 (±6.8)	25.9 (20.9, 31.5)	19.8 (16.6, 23.4)	21.1 (18.5, 24.0)	12.8 (11.1, 14.8)
Gambia	3392	1645	48.5 (43.9, 53.1)	27.9 (±27.9)	50.2 (40.4, 59.9)	17.1 (13.8, 21.1)	53.3 (49.0, 57.4)	20.3 (17.7, 23.2)
Democratic Republic of Congo	7168	3448	48.1 (45.3, 50.9)	27.7 (±7.1)	33.8 (27.9, 40.2)	22.0 (18.4, 26.1)	16.5 (14.5, 18.6)	12.2 (10.5, 14.2)
Niger	5143	2422	47.1 (44.2, 49.9)	27.5 (±10.4)	8.1 (6.2, 10.7)	20.5 (17.8, 23.4)	87.8 (86.1, 89.4)	30.6 (27.7, 33.7)
Sierra Leone	4820	2227	46.2 (42.6, 49.8)	27.3 (±18.8)	31.8 (25.6, 38.7)	19.8 (16.2, 24.0)	62.2 (59.0, 65.3)	19.3 (16.9, 22.0)
Benin	5502	2525	45.9 (43.7, 48.2)	27.8 (±15.8)	39.3 (34.3, 44.5)	20.4 (17.4, 23.6)	58.5 (55.8, 61.1)	15.4 (13.8, 17.2)
Ghana	2264	1005	44.4 (41.4, 47.5)	29.6 (±6.7)	45.7 (38.7, 52.8)	19.0 (15.3, 23.4)	24.8 (21.3, 28.6)	18.3 (15.4, 21.6)
Zimbabwe	2454	1040	42.4 (39.4, 45.4)	26.9 (±9)	30.4 (24.7, 36.7)	19.1 (15.3, 23.6)	1.4 (0.7, 2.7)	17.5 (15.0, 20.3)
Mali	3965	1673	42.2 (39.7, 44.8)	27.5 (±10.7)	18.4 (14.7, 22.8)	20.4 (17.4, 23.8)	81.4 (78.8, 83.7)	13.7 (11.6, 16.1)
Togo	2682	1057	39.4 (36.5, 42.4)	28.7 (±15.1)	35.1 (28.9, 41.7)	20.3 (17.1, 23.9)	42.6 (37.8, 47.5)	20.3 (17.5, 23.4)
Liberia	2650	1028	38.8 (34.9, 42.7)	26.4 (±19.9)	48.8 (39.6, 58.0)	20.0 (16.1,2 4.5)	34.9 (30.8, 39.3)	20.9 (17.7, 24.6)
Lesotho	1369	475	34.7 (31.6, 37.9)	26.3 (±20.5)	27.3 (21.0, 34.8)	16.5 (12.6, 21.4)	0.3 (0.1, 1.3)	17.5 (14.0, 21.7)
Zambia	5074	1735	34.2 (32.1, 36.4)	27.5 (±13.6)	32.3 (27.5, 37.4)	20.8 (18.3, 23.6)	12.4 (10.6, 14.5)	13.6 (11.8, 15.6)
Uganda	5901	2000	33.9 (32.1, 35.8)	26.7 (±8.9)	18.1 (14.7, 22.2)	24.4 (21.6, 27.4)	7.6 (6.5, 9)	25.4 (23.3, 27.6)
South Africa	1386	453	32.7 (29.5, 36.1)	27.6 (±3.6)	58.5 (50.9, 65.7)	20.4 (15.5, 26.3)	1.0 (0.5, 2.2)	18.4 (14.4, 23.3)
Sudan	5622	1760	31.3 (29.3, 33.4)	28.6 (±8.9)	24.6 (20.0, 29.8)	26.9 (21.9, 32.6)	40.9 (36.4, 45.6)	38.0 (34.6, 41.4)
Namibia	1947	561	28.8 (26.6, 31.1)	28.3 (±21.4)	50.1 (43.5, 56.7)	19.3 (15.0, 24.4)	5.1 (3.7, 7.1)	22.0 (18.4, 26.0)
Ethiopia	4308	1150	26.7 (24.4, 29.1)	28.5 (±4.9)	12.4 (8.6, 17.6)	20.8 (16.6, 25.9)	60.2 (55.6, 64.7)	33.1 (28.7, 37.9)
Malawi	6549	1559	23.8 (22.1, 25.5)	26.3 (±13.3)	21.1 (16.0, 27.2)	20.8 (18.1, 23.8)	10.7 (8.7, 13.2)	16.8 (14.1, 19.8)
Rwanda	3236	631	19.5 (18.0, 21.2)	28.6 (±11.5)	19.3 (14.8, 24.8)	19.0 (15.7, 22.8)	11.0 (8.7, 13.8)	23.2 (19.7, 27.0)
Burundi	5412	812	15.0 (13.8, 16.2)	29.2 (±16.4)	13.3 (9.6, 18.2)	21.5 (17.9, 25.5)	40.3 (36.3, 44.5)	22.4 (19.2, 26.0)

**Table 2 ijerph-18-05976-t002:** Prevalence of delayed initiation of breastfeeding among caesarean section births (CSB), vaginal births at home (VBH) and vaginal births at a health facility (VBF).

Region/Country	Sample Size	Prevalence (%) of CSB (95% CI)	Prevalence (%) of Delayed Initiation in CSB (95% CI)	Prevalence (%) of Delayed Initiation In VBF (95% CI)	Prevalence (%) of Delayed Initiation in VBH (95% CI)
East Asia and Pacific					
Thailand	2092	33.4 (29.9, 37.3)	75.8 (70.0, 80.7)	51.5 (45.8, 57.1)	85.5 (63.8, 94.0)
Indonesia	6616	19.2 (17.9, 20.6)	63.2 (59.5, 66.6)	37.3 (35.4, 39.4)	44.1 (39.9, 47.0)
Philippines	3725	15.8 (13.6, 18.1)	62.4 (53.8, 70.3)	38.3 (35.3, 41.4)	45.5 (39.2, 51.9)
Cambodia	2944	8.0 (6.8, 9.5)	74.3 (66.4, 80.8)	31.8 (29.2, 34.6)	51.1 (43.0, 59.1)
Myanmar	1669	21.1 (18.3, 24.1)	44.8 (37.8, 51.9)	26.0 (21.0, 31.7)	31.8 (27.8, 36.0)
Timor Leste	2866	3.5 (2.7, 4.4)	47.1 (36.1, 58.4)	23.7 (20.2, 27.5)	24.3 (21.1, 27.7)
Total for region	19,912	20.2 (19.2, 21.3)	64.0 (61.2, 66.6)	38.8 (37.3, 40.5)	40.4 (37.9, 43.0)
Europe and Central Asia					
Armenia	666	21.4 (18.0, 25.4)	84.6 (75.2, 90.9)	52.1 (46.6, 57.6)	56.9 (14.0, 91.5)
Albania	1035	31.8 (28.1, 35.6)	59.6 (52.3, 66.4)	36.2 (30.6, 42.2)	11.2 (2.5, 38.5)
Moldova	750	16.2 (13.7, 19.4)	79.9 (71.4, 86.4)	30.7 (26.3, 35.4)	68.7 (13.8, 97.2)
Tajikistan	2481	5.9 (4.9, 7.1)	71.4 (62.0, 79.2)	36.5 (32.8, 40.4)	34.5 (27.7, 42.0)
Ukraine	707	12.1 (9.7, 14.9)	70.1 (59.4, 78.9)	28.7 (24.8, 33.0)	85.3 (64.8, 94.8)
Kazakhstan	2157	14.8 (13.0, 16.9)	47.1 (40.4, 53.8)	10.8 (8.8, 13.3)	87.6 (68.5, 95.8)
Kyrgyz Republic	1696	6.9 (5.6, 8.5)	63.9 (51.5, 74.7)	12.5 (10.6, 14.7)	39.1 (12.5, 74.3)
Total for region	9492	12.8 (11.5, 14.2)	64.8 (59.4, 69.8)	25.5 (23.3, 27.9)	54.0 (44.0, 63.6)
Latin America and the Caribbean					
The Dominican Republic	1395	60.6 (56.7, 64.3)	65.5 (60.8, 69.9)	43.7 (37.6, 49.9)	36.6 (18.8, 58.9)
Haiti	2424	5.6 (4.6, 6.8)	81.4 (71.5, 88.4)	49.5 (45.6, 53.4)	51.6 (48.3, 54.8)
Guyana	769	17.0 (14.5, 20.0)	77.2 (68.5, 84.1)	44.2 (40.0, 48.4)	57.7 (46.1, 66.8)
Guatemala	4790	29.6 (27.6, 31.6)	69.0 (65.9, 72.1)	27.4 (24.9, 30.1)	18.3 (15.5, 21.3)
Total for region	9378	31.0 (29.0, 32.9)	67.9 (65.1, 70.6)	37.8 (35.5, 40.2)	36.8 (34.2, 39.5)
Middle East and North Africa					
Egypt	6297	57.4 (55.5, 59.2)	82.3 (80.4, 84.0)	63.2 (60.4, 66.0)	52.2 (47.8, 56.6)
Tunisia	1164	26.7 (23.7, 29.9)	84.8 (78.8, 89.3)	50.9 (45.9, 55.8)	59.7 (38.9, 77.4)
Yemen	6110	5.7 (4.9, 6.6)	78.4 (72.3, 83.5)	49.5 (46.2, 52.7)	43.9 (41.5, 46.3)
Jordan	3472	28.2 (26.0, 30.6)	52.2 (47.4, 56.9)	25.8 (23.1, 28.7)	18.4 (3.0, 62.3)
Total for region	17,043	43.1 (41.5, 44.7)	81.0 (79.3, 82.6)	54.5 (52.4, 56.6)	47.0 (44.7, 49.3)
South Asia					
Pakistan	3935	25.8 (23.1, 28.6)	91.4 (88.5, 93.6)	76.9 (73.5, 80.0)	76.2 (71.5, 80.4)
Afghanistan	11,539	3.6 (2.9, 4.5)	76.7 (67.1, 84.1)	57.1 (54.6, 59.6)	60.0 (56.1, 63.7)
India	94,111	19.1 (18.7, 19.6)	66.5 (65.3, 67.7)	54.1 (53.5, 54.8)	65.5 (64.4, 66.6)
Bangladesh	3205	24.6 (22.2, 27.3)	71.2 (67.0, 75.0)	45.8 (40.2, 51.6)	41.1 (37.7, 44.4)
Nepal	1978	10.0 (8.3, 12.0)	77.5 (70.1, 83.5)	34.6 (31.2, 38.1)	52.6 (47.0, 58.1)
The Maldives	1086	43.0 (39.6, 46.5)	34.5 (29.1, 40.3)	33.1 (27.8, 38.8)	0
Total for region	115,854	20.0 (19.4, 20.5)	71.0 (69.8, 72.2)	55.9 (55.1, 56.6)	61.0 (59.4, 62.5)
Sub-Saharan Africa					
Guinea	2818	3.0 (2.2, 3.9)	92.0 (79.1, 97.2)	80.8 (76.8, 84.2)	84.7 (81.3, 87.6)
Chad	6742	1.5 (1.2, 2.0)	92.9 (84.1, 97.0)	77.1 (73.4, 80.4)	76.6 (73.9, 79.2)
Cote d’Ivoire	3039	3.0 (2.3, 3.9)	87.0 (77.3, 93.0)	67.6 (63.7, 71.3)	70.1 (65.0, 74.7)
Cameroon	2977	2.6 (2.0, 3.3)	74.8 (63.0, 83.8)	65.5 (62.3, 68.6)	73.4 (68.9, 76.7)
Gabon	2102	10.6 (8.3, 13.5)	89.4 (79.7, 94.7)	65.9 (61.1, 70.3)	53.2 (44.0, 62.3)
Nigeria	12,473	2.2 (1.9, 2.6)	79.7 (73.5, 84.8)	58.4 (56.1, 60.7)	70.9 (68.6, 73.0)
Senegal	4447	5.7 (4.8, 6.8)	95.1 (91.5, 97.2)	61.1 (58.6, 63.7)	79.2 (75.0, 82.9)
Comoros	1298	11.4 (9.2, 13.9)	82.4 (71.3, 89.9)	63.3 (58.8, 67.6)	66.9 (57.8, 74.9)
Sierra Leone	4820	4.0 (3.3, 4.9)	61.4 (50.8, 71.0)	49.5 (45.3, 53.6)	40.2 (35.6, 45.0)
Sao Tome	756	5.6 (3.7, 8.4)	92.1 (77.7, 97.5)	60.6 (55.2, 65.7)	54.3 (41.4, 64.3)
Angola	5405	4.0 (3.2, 4.9)	74.0 (63.2, 82.5)	47.6 (44.1, 51.1)	53.6 (50.2, 57.1)
Gambia	3392	2.0 (1.4, 2.8)	66.1 (46.8, 81.1)	50.0 (45.3, 54.6)	44.7 (38.1, 51.6)
Democratic Republic of Congo	7168	5.3 (4.4, 6.3)	76.0 (68.4, 82.3)	44.2 (41.1, 47.3)	56.0 (51.2, 60.6)
Niger	5143	1.4 (1.1, 1.9)	63.6 (50.1, 75.2)	30.5 (26.9, 34.3)	55.1 (51.7, 58.4)
Benin	5502	5.0 (4.4, 5.8)	75.1 (68.8, 80.4)	44.8 (42.4, 47.3)	41.6 (36.8, 46.6)
Ghana	2264	12.5 (10.7, 14.6)	71.9 (63.5, 78.9)	37.9 (34.0, 41.8)	47.0 (41.9, 52.1)
Zimbabwe	2454	6.2 (5.1, 7.4)	79.3 (72.1, 85.0)	36.6 (33.6, 39.7)	56.1 (48.6, 63.3)
Mali	3965	3.0 (2.4, 3.7)	62.8 (52.8, 71.9)	38.8 (35.6, 42.0)	45.6 (41.8, 49.4)
Togo	2682	7.5 (6.2, 8.9)	62.9 (54.5, 70.6)	33.5 (30.2, 36.9)	48.5 (43.3, 53.8)
Liberia	2650	4.5 (3.4, 5.8)	74.1 (62.9, 82.8)	36.4 (32.2, 40.8)	38.1 (32.5, 44.0)
Lesotho	1369	10.2 (8.5, 12.0)	60.3 (50.2, 69.6)	32.6 (29.0, 36.4)	28.7 (23.0, 35.3)
Zambia	5074	4.6 (3.9, 5.3)	60.7 (53.1, 67.9)	29.4 (27.2, 31.8)	42.1 (38.1, 46.2)
Uganda	5901	7.3 (6.4, 8.3)	63.2 (57.5, 68.5)	29.2 (27.1, 31.3)	39.3 (36.1, 42.7)
South Africa	1386	24.8 (21.8, 28.0)	40.9 (35.0, 47.0)	29.8 (26.1, 33.7)	35.2 (19.4, 55.0)
Tanzania	4076	6.6 (5.6, 7.8)	85.0 (78.7, 89.7)	38.3 (35.6, 41.1)	60.1 (55.5, 64.6)
Sudan	5622	6.1 (5.2, 7.2)	54.3 (47.1, 61.5)	31.5 (28.2, 35.1)	29.3 (26.7, 31.5)
Namibia	1947	15.7 (13.7, 17.8)	47.6 (40.7, 54.6)	25.6 (23.1, 28.2)	23.1 (17.9, 29.2)
Ethiopia	4308	2.8 (2.1, 3.7)	62.7 (48.8, 74.7)	24.2 (20.9, 27.8)	26.5 (23.6, 29.5)
Malawi	6549	6.6 (5.8, 7.4)	48.6 (42.1, 55.1)	21.3 (19.8, 23.0)	32.0 (26.2, 38.5)
Rwanda	3236	13.4 (12.1, 14.8)	56.6 (51.7, 61.5)	12.4 (11.0, 14.0)	30.2 (23.8, 37.5)
Burundi	5412	5.4 (4.7, 6.2)	64.6 (58.1, 70.7)	11.7 (10.6, 13.0)	15.8 (12.2, 20.2)
Total for region	126,977	5.8 (5.5, 6.1)	62.9 (60.4, 65.3)	40.9 (40.0, 41.8)	53.9 (52.3, 55.5)

**Table 3 ijerph-18-05976-t003:** Population Attributable Risk percent for vaginal births at home (VBH) and caesarean section births (CSB) relative to vaginal births at a health facility (VBF).

Region/Country	Crude PAR% for VBH (95% CI)	Crude PAR% for CSB (95% CI)	Adjusted PAR for VBH (95% CI) **	Adjusted PAR% for CSB (95% CI) **
East Asia and Pacific				
Thailand	32.6% (16.2, 47.2)	24.3% (16.9, 31.5)	4.3% (−25.7, 33.6)	20.7% (13, 28.2)
Indonesia	6.1% (2.1, 10.0)	25.8% (21.8, 29.7)	3.4% (−1.5, 8.3)	27.0% (22.8, 31.0)
Philippines ^‡^	7.2% (0.3, 14.0)	24.1% (14.8, 33.0)	-	-
Cambodia	19.2% (11.0, 27.2)	42.4% (34.6, 49.7)	15.6% (5.2, 25.7)	33.3% (25.5, 40.8)
Myanmar	5.8% (−0.7, 12.1)	18.7% (9.9, 27.2)	−3.1% (−12.5, 6.4)	18.2% (9.2, 26.8)
Timor Leste	0.6% (−3.7, 4.8)	23.4% (11.3, 34.9)	−2.0% (−10.1, 6.0)	19.8% (9.3, 29.9)
Europe and Central Asia				
Armenia	4.8% (−39.5, 47.3)	32.5% (23.3, 41.1)	5.3% (−40.8, 49.2)	29.6% (19.8, 38.8)
Albania	−25.0% (−43.4, −4.6)	23.4% (14.5, 31.9)	−26.6% (−40.7, −11.1)	23.0% (14.7, 31.0)
Moldova	39.6% (−19.9, 77.7)	49.2% (40.0, 57.4)	22.9% (−48.1, 75.8)	46.2% (35.9, 55.5)
Tajikistan	−2.0% (−9.5, 5.5)	34.9% (25.3, 43.8)	−1.9% (−9.7, 5.9)	35.9% (26.1, 44.9)
Ukraine	56.6% (39.0, 70.3)	41.4% (30.7, 51.0)	−28.5% (−32.4, −24.6)	40.1% (28.5, 50.5)
Kazakhstan	76.8% (54.6, 88.9)	36.2% (28.5, 43.5)	21.5% (−36.2, 67.3)	36.7% (29.0, 43.9)
Kyrgyz Republic	26.6% (−10.7, 57.3)	51.4% (38.8, 62.1)	−2.0% (−19.2, 15.2)	45.8% (30.6, 58.7)
Latin America and the Caribbean				
The Dominican Republic	−7.1% (−30.7, 17.3)	21.8% (15.3, 28.2)	11.0% (−29.3, 47.9)	20.4% (13.4, 27.2)
Haiti	2.1% (−2.7, 6.8)	31.9% (22.0, 41.1)	7.3% (−3.9, 18.4)	33.0% (23.1, 42.2)
Guyana	12.5% (1.3, 23.5)	33.1% (24.5, 41.1)	3.3% (−20.7, 26.9)	27.4% (18.4, 36.0)
Guatemala	−9.2% (−13.0, −5.3)	41.6% (37.8, 45.3)	−6.2% (−10.8, −1.6)	38.9% (35.0, 42.7)
The Middle East and North Africa				
Egypt	−11.0% (−16.2, −5.8)	19.0% (16.0, 22.1)	−9.2% (−17.1, −1.1)	18.3% (15.1, 21.4)
Tunisia	8.8% (−12.3, 29.1)	33.9% (26.2, 41.2)	7.5% (−17.1, 31.3)	30.5% (22.3, 38.2)
Yemen^ǂ^	−5.6% (−9.3, −1.9)	28.9% (22.9, 34.8)	-	-
Jordan	−7.4% (−35.5, 21.9)	26.4% (21.1, 31.5)	1.1% (−37.1, 39)	25.9% (20.6, 31.1)
South Asia				
Pakistan	−0.6% (−5.5, 4.3)	14.5% (11.0, 18.0)	−0.8% (−7.5, 6.0)	14.0% (10.4, 17.6)
Afghanistan	2.9% (−1.3, 7.0)	19.6% (11.2, 27.7)	0.0% (−5.9, 6.0)	18.3% (9.7, 26.7)
India	11.4% (10.2, 12.6)	12.4% (11.0, 13.7)	3.9% (2.5, 5.4)	15.7% (14.2, 17.2)
Bangladesh	−4.8% (−11.0, 1.4)	25.3% (18.2, 32.2)	−0.3% (−8.8, 8.3)	26.3% (19.3, 33.1)
Nepal	18.0% (11.6, 24.2)	42.9% (35.0, 50.2)	14.5% (5.1, 23.6)	41.4% (33.2, 48.9)
The Maldives ^‡^	−33.1% (−38.3, −27.7)	1.4% (−6.0, 8.7)	-	-
Sub-Saharan Africa				
Guinea	3.9% (−0.1, 7.9)	11.2% (3.2, 19.2)	7.6% (1.0, 14.0)	10.3% (2.0, 18.5)
Chad	−0.4% (−4.2, 3.4)	15.9% (9.2, 22.5)	2.0% (−3.6, 7.6)	17.5% (10.0, 24.8)
Cote d’Ivoire	2.5% (−2.4, 7.3)	19.4% (10.7, 27.8)	5.9% (−6.0, 17.6)	18.6% (9.5, 27.4)
Cameroon	7.4% (2.9, 11.9)	9.2% (−1.4, 19.6)	−2.0% (−10.9, 6.8)	12.7% (2.3, 22.9)
Gabon	−12.6% (−22.4, −2.6)	23.5% (15.4, 31.3)	−20.5% (−30.7, −9.9)	20.0% (10.9, 28.8)
Nigeria	12.5% (9.8, 15.2)	21.3% (15.3, 27.2)	6.4% (2.0, 10.8)	20.8% (15.1, 26.4)
Senegal	18.1% (13.9, 22.2)	33.9% (30.3, 37.4)	17.1% (12.1, 22.1)	33.1% (28.4, 37.7)
Comoros	3.6% (−5.4, 12.5)	19.1% (9.8, 28.2)	1.3% (−11.5, 14.0)	20.1% (11.2, 28.7)
Sao Tome	−7.6% (−19.0, 4.0)	31.5% (22.2, 40.3)	−24.8% (−37.3, −11.4)	28.4% (18.1, 38.0)
Sierra Leone	−9.3% (−14.5, −4.0)	11.9% (1.9, 21.7)	−7.0% (−13.5, −0.3)	11.5% (1.8, 21.0)
Angola	6.1% (1.8, 10.3)	26.4% (16, 36.3)	−4.4% (−12.8, 4.1)	25.7% (15.0, 35.9)
Gambia	−5.3% (−11.7, 1.2)	16.1% (−1.4, 32.6)	−5.1% (−19.9, 10.0)	14.0% (−2.1, 29.4)
Democratic Republic of Congo	11.8% (6.9, 16.7)	31.9% (24.5, 38.9)	12.9% (5.3, 20.4)	30.7% (23.1, 37.9)
Niger	24.6% (20.0, 29.0)	33.1% (19.3, 45.6)	4.7% (−4.9, 14.2)	38.2% (22.3, 52.1)
Benin	−3.2% (−8.3, 1.8)	30.2% (23.9, 36.4)	−0.7% (−7.7, 6.4)	27.3% (20.6, 33.7)
Ghana	9.1% (3.2, 14.9)	34.0% (24.6, 42.7)	−12.9% (−27.6, 2.4)	32.5% (22.6, 41.8)
Zimbabwe	19.5% (12.0, 26.8)	42.7% (35.3, 49.5)	−3.7% (−12.3, 4.9)	41.3% (32.5, 49.4)
Mali	6.8% (2.0, 11.6)	24.1% (13.9, 33.7)	1.1% (−6.9, 9.1)	27.0% (16.7, 36.8)
Togo	15.0% (9.2, 20.8)	29.5% (20.9, 37.6)	6.0% (−1.4, 13.4)	33.1% (23.4, 42.2)
Liberia	1.7% (−4.4, 7.7)	37.7% (27.0, 47.4)	4.8% (−4.7, 14.3)	33.5% (22.3, 43.8)
Lesotho	−3.9% (−10.7, 2.9)	27.7% (16.8, 37.9)	−15% (−28.1, −1.4)	24.3% (13.3, 34.7)
Zambia	12.6% (8.5, 16.8)	31.3% (23.6, 38.6)	0.6% (−5.6, 6.8)	30.8% (22.9, 38.4)
Uganda	10.1% (6.5, 13.8)	34.0% (28.2, 39.5)	−0.3% (−5.3, 4.7)	38.8% (32.2, 45.1)
South Africa	5.4% (−13.2, 23.7)	11.1% (3.9, 18.2)	−1.4% (−21.5, 18.8)	10.6% (3.3, 17.7)
Tanzania	21.8% (17.3, 26.3)	46.7% (40.6, 52.4)	20.1% (15.1, 24.9)	47.5% (39.6, 54.7)
Sudan	−2.5% (−6.6, 1.5)	22.8% (14.8, 30.5)	−9.9% (−15.1, −4.6)	23.2% (15.3, 30.8)
Namibia	−2.5% (−8.5, 3.5)	22.0% (14.2, 29.5)	−7.7% (−19.2, 4.0)	21.1% (13.0, 28.9)
Ethiopia	2.3% (−2.3, 6.8)	38.5% (24.3, 51.0)	−2.3% (−13.4, 8.8)	40.6% (22.2, 56.2)
Malawi	10.7% (4.3, 16.9)	27.2% (20.6, 33.7)	7.4% (−0.5, 15.2)	24.0% (17.3, 30.4)
Rwanda	17.8% (10.9, 24.6)	44.2% (39.1, 49.1)	10.8% (−0.4, 21.8)	41.3% (35.7, 46.6)
Burundi	4.1% (0.2, 8.0)	52.9% (46.1, 59.1)	−4.1% (−7.4, −0.7)	54.8% (46.8, 62.0)

Reference group for calculating Population Attributable Risk % (PAR%): vaginal birth at a facility (VBF). ** Adjusted for the size of child at birth, place of residence, age of mother, parity, skilled assistance at birth, number of antenatal visits, the skill level of antenatal care provider, mother’s education, number of antenatal visits, and household wealth index. ^‡^ Multivariable Poisson model could not converge as estimates for some adjusting variables did not exist in the dataset.

## Data Availability

Data used in this study is publicly available through the DHS website (https://www.dhsprogram.com/data (accessed on 30 April 2019)) and is de-identified for anonymity.

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
