# Peer review of "Delayed Initiation of Breastfeeding and Role of Mode and Place of Childbirth: Evidence from Health Surveys in 58 Low- and Middle- Income Countries (2012–2017)"

_ijerph, 2021, doi:10.3390/ijerph18115976_

Round 1
Reviewer 1 Report
This study described the recent country-level prevalence of delayed breastfeeding initiation by place and mode of childbirth in 58 LMIC countries using publicly available survey data. The paper is interesting and overall well written. I recommend that a minor revision is warranted so to clarify some points.
- Introduction: Harms for the delaying breastfeeding initiation can be briefly described and talk more about the innovation of the study.
- Introduction: Line 98-103, this study investigated the influences of place of birth and mode of birth on the delaying breastfeeding initiation. More results of the previous studies are suggested.
- Line 168: the outcomes should be clearly defined.
- Results: Are the data of “Prevalence of delayed initiation of breastfeeding” in table 1 and in that of figure 1 presented the same results? If yes, I suggest to choose one of them.
Author Response
- Introduction: Harms for the delaying breastfeeding initiation can be briefly described and talk more about the innovation of the study.
Response: Thank you for the suggestion. We have reshuffled some of the relevant text to bring all harmful effects of delayed initiation into the paragraph from Line 78-97. We have also added more text and references from studies reporting the harms of delaying breastfeeding initiation. The added text reads-
“A review of 18 studies found delayed initiation of breastfeeding reduced the pooled risk of all-cause mortality by 44% among newborns who survived past 48 hours after birth and by 42% among low birth weight infants. Initiating breastfeeding between 2 and 23 hours after birth is associated with a 33% increased risk of neonatal death compared to breastfeeding within that first hour of birth.”
We have also modified the text on the uniqueness of the study in Line 137-140. The modified text reads -
“Unlike previous studies, the unique aspect of our analysis is that we examine the proportion of delayed initiation of breastfeeding that could be averted if all women had the same risk of delaying initiation as those who experience a vaginal birth in a health facility.”
- Introduction: Line 98-103, this study investigated the influences of place of birth and mode of birth on the delaying breastfeeding initiation. More results of the previous studies are suggested.
Response: Thank you for the suggestion. We have referred to 11 previous studies that have established the relationship between the time of breastfeeding initiation and the place and mode of childbirth separately [Line 119-130]. We agree that the text referring to these studies did not strongly establish the association, and so we have modified the text and the information as follows:
“Most evidence suggests that early breastfeeding initiation is higher among women experiencing hospital births in LMICs [4,27-30,32,33]. However, the findings are not consistent for all hospital births. Studies in Nigeria [26], Ethiopia [31], Uganda [34], and India [35] report that mothers who experienced a caesarean section birth at a health facility had a significantly higher likelihood of delaying initiation of breastfeeding beyond the first hour of birth, compared to those experienced vaginal births.”
- Line 168: the outcomes should be clearly defined.
Response: We think we have clearly defined the outcomes in Line 203-211. We do not see any need for further clarification of the outcome, and therefore, we have made no changes to the manuscript.
- Results: Are the data of “Prevalence of delayed initiation of breastfeeding” in table 1 and in that of figure 1 presented the same results? If yes, I suggest to choose one of them.
Response: Thank you for picking up this duplicated data. The results presented in the column “Prevalence of delayed initiation of breastfeeding” in Table 1 and Figure 1 are the same. Table 1 includes the confidence intervals of the country-level prevalence, and we believe it is more informative. We have decided to put the figure in the supplementary materials so readers can also visualise the distribution of the prevalence of delayed breastfeeding initiation.

Reviewer 2 Report
This manuscript describes an analysis of the effect of mode and place of delivery on delayed breastfeeding initiation in 58 low and middle income countries. This is an important area of research and would be of interest to those working in this area. However, there are some areas which require clarification.
The authors state the cut of used to define delayed feeding was either within or after 1 hour of birth. Where would women who have stated 1 hour have been classified (above or below). Cut offs can be a rather rigid tool, if it is 59 mins vs 60 mins does it really matter (and can a women give this level of precision retrospectively - especially with the drama of childbirth). But if it 59 minutes vs 3 hours then this does matter. The precision of the data collected an the use of a broad brush cut off are limitations of the study which should be acknowledged.
Could the authors provide the mean/median for the time for those over 1 hour after delivery? It would also be good to show the range to give an idea of the spread of the data? this could show if women were just over the 1 hour or were many hours over the hour, which is important if we are to address practice. Is it a mountain to climb or a small hill.
Author Response
This manuscript describes an analysis of the effect of mode and place of delivery on delayed breastfeeding initiation in 58 low and middle income countries. This is an important area of research and would be of interest to those working in this area. However, there are some areas which require clarification.
- The authors state the cut of used to define delayed feeding was either within or after 1 hour of birth. Where would women who have stated 1 hour have been classified (above or below). Cut offs can be a rather rigid tool, if it is 59 mins vs 60 mins does it really matter (and can a women give this level of precision retrospectively - especially with the drama of childbirth). But if it 59 minutes vs 3 hours then this does matter. The precision of the data collected an the use of a broad brush cut off are limitations of the study which should be acknowledged.
Could the authors provide the mean/median for the time for those over 1 hour after delivery? It would also be good to show the range to give an idea of the spread of the data? this could show if women were just over the 1 hour or were many hours over the hour, which is important if we are to address practice. Is it a mountain to climb or a small hill.
Response: Thank you for raising this very important matter. The data collected by the Demographic and Health Surveys and Multiple Indicator Cluster Survey Does not collect precise information on the exact time of breastfeeding initiation in minutes. The DHS and MICS collect this data as i) ‘immediately’ for those who initiated within the first hour of birth, ii) in ‘hours’ if the woman initiated breastfeeding after the first hour but within the first day, and iii) in ‘days’ if initiation was after the first day. We have added the median time of initiation (for all countries combined) for those who initiated beyond the first hour of birth. The added text reads as follows:
“In all countries, the overall median time to initiate breastfeeding, among those who initiated after the first hour of birth, was 2 hours (interquartile range- 47 hours).”
To address the distribution of increased delay across countries, we have included a supplementary table with the median time to initiation of breastfeeding among those who initiated after the first hour of birth. We have also included the interquartile range, 25th and 75th percentiles, and the minimum and maximum delay in breastfeeding initiation.
|
Supplementary Table 3: Median time (in hours) of initiation of breastfeeding among delayed initiators* |
||||||
|
Region/Country |
Median time to breastfeeding initiation (hrs) |
Interquartile range (hrs) |
25th percentile (hrs) |
75th percentile (hrs) |
Minimum delay (hrs) |
Maximum delay (hrs) |
|
East Asia & Pacific |
||||||
|
Thailand |
3 |
47 |
1 |
48 |
1 |
2184 |
|
Indonesia |
48 |
70 |
2 |
72 |
1 |
576 |
|
Philippines |
2 |
11 |
1 |
12 |
1 |
576 |
|
Cambodia |
2 |
11 |
1 |
12 |
1 |
408 |
|
Myanmar |
6 |
71 |
1 |
72 |
1 |
576 |
|
Timor Leste |
1 |
2 |
1 |
3 |
1 |
360 |
|
Europe & Central Asia |
||||||
|
Armenia |
3 |
5 |
2 |
7 |
1 |
384 |
|
Albania |
2 |
4 |
1 |
5 |
1 |
384 |
|
Moldova |
3 |
47 |
1 |
48 |
1 |
528 |
|
Tajikistan |
2 |
2 |
1 |
3 |
1 |
504 |
|
Ukraine |
3 |
8 |
2 |
10 |
1 |
864 |
|
Kazakhstan |
3 |
46 |
2 |
48 |
1 |
1464 |
|
Kyrgyz Republic |
2 |
15 |
1 |
16 |
1 |
504 |
|
Latin America & Caribbean |
||||||
|
Dominican Republic |
8 |
46 |
2 |
48 |
1 |
408 |
|
Haiti |
2 |
11 |
1 |
12 |
1 |
576 |
|
Guyana |
3 |
47 |
1 |
48 |
1 |
1464 |
|
Guatemala |
5 |
46 |
2 |
48 |
1 |
744 |
|
Middle East & North Africa |
||||||
|
Egypt |
3 |
8 |
2 |
10 |
1 |
384 |
|
Tunisia |
4 |
46 |
2 |
48 |
1 |
744 |
|
Yemen |
5 |
70 |
2 |
72 |
1 |
720 |
|
Jordan |
5 |
46 |
2 |
48 |
1 |
528 |
|
South Asia |
||||||
|
Pakistan |
6 |
70 |
2 |
72 |
1 |
744 |
|
Afghanistan |
2 |
3 |
1 |
4 |
1 |
384 |
|
India |
2 |
11 |
1 |
12 |
1 |
768 |
|
Bangladesh |
2 |
4 |
1 |
5 |
1 |
528 |
|
Nepal |
2 |
5 |
1 |
6 |
1 |
576 |
|
Maldives |
1 |
7 |
1 |
8 |
1 |
576 |
|
Sub-Saharan Africa |
||||||
|
Guinea |
4 |
46 |
2 |
48 |
1 |
360 |
|
Chad |
72 |
90 |
6 |
96 |
1 |
504 |
|
Cote d'Ivoire |
7 |
45 |
3 |
48 |
1 |
744 |
|
Cameroon |
4 |
46 |
2 |
48 |
1 |
696 |
|
Gabon |
3 |
46 |
2 |
48 |
1 |
576 |
|
Nigeria |
5 |
46 |
2 |
48 |
1 |
576 |
|
Senegal |
3 |
5 |
1 |
6 |
1 |
576 |
|
Comoros |
3 |
7 |
2 |
9 |
1 |
264 |
|
Sierra Leone |
3 |
5 |
1 |
6 |
1 |
360 |
|
Sao Tome |
2 |
4 |
1 |
5 |
1 |
504 |
|
Angola |
2 |
6 |
1 |
7 |
1 |
576 |
|
Gambia |
2 |
3 |
1 |
4 |
1 |
480 |
|
Democratic Republic of Congo |
2 |
5 |
1 |
6 |
1 |
384 |
|
Niger |
4 |
47 |
1 |
48 |
1 |
576 |
|
Benin |
3 |
47 |
1 |
48 |
1 |
576 |
|
Ghana |
3 |
15.5 |
1 |
16.5 |
1 |
528 |
|
Zimbabwe |
2 |
5 |
1 |
6 |
1 |
576 |
|
Mali |
1 |
2 |
1 |
3 |
1 |
504 |
|
Togo |
2 |
47 |
1 |
48 |
1 |
744 |
|
Liberia |
3 |
47 |
1 |
48 |
1 |
360 |
|
Lesotho |
4 |
46 |
2 |
48 |
1 |
576 |
|
Zambia |
2 |
3 |
1 |
4 |
1 |
576 |
|
Uganda |
2 |
3 |
1 |
4 |
1 |
576 |
|
South Africa |
3 |
11 |
1 |
12 |
1 |
360 |
|
Tanzania |
2 |
5 |
1 |
6 |
1 |
528 |
|
Sudan |
2 |
47 |
1 |
48 |
1 |
984 |
|
Namibia |
2 |
47 |
1 |
48 |
1 |
528 |
|
Ethiopia |
3 |
47 |
1 |
48 |
1 |
576 |
|
Malawi |
2 |
3 |
1 |
4 |
1 |
576 |
|
Rwanda |
2 |
5 |
1 |
6 |
1 |
744 |
|
Burundi |
1 |
2 |
1 |
3 |
1 |
576 |
|
*Demographic and Health Surveys (DHS) and Multiple Indicator Cluster Survey (MICS) does not collect precise information on the exact time of breastfeeding initiation. The data is collected as i) ‘immediately’ for those who initiated within the first hour of birth, ii) in ‘hours’ if breastfeeding was initiated after the first hour but within the first day, and iii) in ‘days’ if initiation was after the first day. |
||||||
